# Green Care Achievement Based on Aquaponics Combined with Human–Computer Interaction

**Wei-Ling Lin** [1] , **Shu-Ching Wang** [2] , **Li-Syuan Chen** [3,*], **Tzu-Ling Lin** [4] **and Jian-Le Lee** [5]

1   Bachelor Degree Program of Artificial Intelligence, National Taichung University of Science and Technology, Taichung City 404, Taiwan
2   Department of Information Management, Chaoyang University of Technology, Taichung City 413, Taiwan
3   Department of Computer Science and Engineering, National Chung Hsing University, Taichung City 402, Taiwan
4   Department of Business Management, National Taichung University of Science and Technology, Taichung City 404, Taiwan
5   Office of Academic Affairs, National Taichung University of Science and Technology, Taichung City 404, Taiwan
*   Correspondence: d110056006@mail.nchu.edu.tw

**Abstract:** According to the "World Population Prospects 2022" released by the United Nations in August 2022, the world will officially enter an "aging society". In order to provide the elderly with an improved quality of daily life, "health promotion" and "prevention of disease" will be important. With respect to care of the elderly, the concepts of "therapeutic environment" and "green care" have been explored and developed. Therefore, in this study, we combine the currently popular Internet of Things (IoT) into an aquaponics system and proposes a smart green care system (SGCS). The proposed system uses face recognition technology to record the labor and rehabilitation history of the elderly, in combination with environmental data analysis, to enable automatic control decisions for equipment in conjunction with a voice control system to reduce the obstacles faced by the elderly in operating the information system. It also uses image recognition technology to monitor and notify about plant diseases and insect pests to achieve automatic management and enhance the interaction between the elderly and the SGCS through human–computer interaction. The SGCS allows the elderly to guide it to participate in appropriate activities through direct contact with the natural environment, thereby enhancing the quality of green healing life. In this study, taking long-term care institutions as an example, we verified proof of concept (PoC), proof of service (PoS), and proof of business (PoB), confirming the feasibility of the SGCS. The SGCS proposed in this study can be successfully used in long-term care institutions and various other environments, such as medical units and home care contexts. It can take full advantage of the functions associated with the concept of "healing environment" and "green care" widely recognized by users. Therefore, it can be widely used in the field of long-term care in the future.

**Keywords:** green care; Internet of Things; smart green care system; therapeutic environment

## 1. Introduction

The United Nations has long been observing global demographic trends and predicting future demographic changes. According to the World Population Prospects 2022 report, the world's population will continue to grow, reaching 7.7 billion in 2022. The demographic structure is also noticeably changing. The world's population will reach 9.8 billion within the next 30 years, with 1.5 billion people over the age of 65, representing more than 16% of the global population. Therefore, the global population is transforming into a super-aged society [1]. Due to the reduced mobility of elderly and chronic patients, their scope of life and social interaction is limited, resulting in physical and mental impacts, aggravating the extent of physical aging. Society must provide more medical care, more

effective disease prevention, and a friendlier environment so that the elderly are not confined to their own houses. It would be helpful for the elderly to enter society and expand their social circles. Therefore, it is particularly important to promote healthy living among the current elderly population. To establish a medical and caring environment suitable for this population, a series of studies has been conducted related to the theme of therapeutic environment. The authors of these studies have made suggestions to improve environmental software and hardware, with the aim of creating a healthy, happy, and friendly therapeutic environment. The plan involves providing a therapeutic environment to care for the elderly, the intellectually disabled, the chronically ill, and the physically and mentally handicapped. The proposed environment would represent a suitable living space for these populations to improve their physical and mental health and help them gain peace of mind and body [2]. Green care is a form of conceptual nursing care promoted by the Scientific and Technological Cooperation Committee (COST), a multinational coordinating unit established by the European Union. It distinguishes between the experience of the natural environment and interaction with natural elements and has three healthcare goals: health promotion, treatment, and labor generation. The aim of green care is to integrate the concept of health care through agriculture, horticulture, animals, and natural spaces and to provide a multifunctional treatment for the elderly, dementia patients, the chronically ill, the mentally handicapped, etc. To date, a multifaceted treatment plan has been developed, including animal-assisted interventions, horticultural therapy, care farming, green exercise, ecotherapy, nature therapy, wildlife therapy, etc., [3].

At this stage, whether it is a therapeutic environment or green care, the plan is suited to a large-scale institutional care environment. With respect to home care, such an environment cannot be constructed. At present, these two types of software and hardware environments are also suited to spatial planning and lack the concept of physical and spiritual care of the care recipients. Therefore, with this study, we hope to create a care environment that is suitable for institutional and home use by using information technology to interact with the care recipients and improve their physical and mental health.

Aquaponics is also known as composite farming systems, i.e., a combination of aquaculture and mutual symbiosis of hydroponic ecosystems, which can be cultivated as soil through a medium, as well as pure or tubular farming, requiring little water and land, making such a system ideal for the top floors or balconies in current metropoles. Such a system can produce fresh fruits and vegetables independent of the influence of climatic conditions and with much lower energy consumption than traditional agriculture [4]. Most of today's aquaponics systems combine Internet of Things (IoT) technology to detect current water quality and soil conditions through the deployment of IoT technology and sensors [5,6]. However, most of today's agricultural Internet of Things applications use sensors to obtain environmental data for user reference, with remote control to operate equipment. Few of these systems incorporate data use and analysis, lacking the concept of big data analysis. Therefore, in this research, we hopes to combine the IoT, image analysis, and big data analysis to enable decision making for remote equipment control, as well as to monitor and report plant diseases and insect pests in order to mitigate dilemma of system maintenance and operation.

The results of this study can be used to promote effective utilization of environmental resources through aquaponics systems for the development of an environment that incorporates the concept of green care to enable green treatment and care for the elderly, the intellectually disabled, the chronically ill and the mentally handicapped, integrating human–computer interaction design with the proposed SGCS. The elderly population is encouraged to engage with the natural environment to perform physical care and rehabilitation in order to achieve the goals of maintaining physical health and providing psychological comfort. In addition, to solve the problem of the elderly's inability to operate information interfaces, the intelligent voice control system (IVCS) proposed by Wang and others [7] is integrated into the design of the system proposed in this study. The system

uses mobile devices to communicate directly via natural language to achieve automatic voice control and guided decision control.

The main contribution of the proposed SGCS is that it can be used to cost-effectively develop a popular and stable care environment, improving the physical and mental health of care recipients at home reducing pressure on caregivers.

The remainder of this article is arranged as follows. In Section 2, we review the literature. In Section 3, we discuss the research concept and the proposed SGCS. In Section 4, we discuss and analyze the effectiveness of the proposed SGCS. Finally, in Section 5, we outline our conclusions.

## 2. Related Work

In this section, the concepts of therapeutic environment and green care, aquaponics systems, cloud computing, and the Internet of Things will be discussed.

### 2.1. Therapeutic Environment and Green Care

World Population Prospects 2022 [1] predicts future population trends and provides relevant policy references for countries with a view to alleviate challenges. In 2022, the world's population reached 7.7 billion people, with a steady growth trend. By 2050, the world's population will reach 9.8 billion, with 1.5 billion people over the age of 65, constitute 16% of the total population. The world is on the threshold of becoming an ultra-old-age society, facing the harsh challenges of a population crisis. Countries have put forward policies to encourage childbirth, delay retirement age, promote advanced education, and set up a comprehensive long-term care system, showing that governments are facing the impact and concerns of an aging population and the problems associated with a declining birthrate [8].

"Health promotion" and "prevention of disease" in the daily lives of the elderly are important concepts. With respect to elder care, the concepts of therapeutic environment and green care have been explored and developed. The so-called therapeutic environment refers the provision of a special environment incorporating the concepts of sociality, individualization, and stability to relieve the pressure of the elderly population. Through space planning, a natural landscape that provides physiological and spiritual pleasure can be designed. Green care integrates agriculture, horticulture, animals, and other natural scenery into the concept of health care. Connection with elements of nature is used as a treatment method to promote physical, psychological, social, and educational well-being and improve quality of life. Combining a therapeutic environment and green care can make people feel happy about their living environment, reduce psychological pressure, and increase comfort [9].

In terms health care for the elderly, there are still many problems to be solved, and several studies point to the hope of building cloud-based medical services [10–14] to solve the problem of remote care and nursing manpower shortages in remote rural areas. With respect to historical nursing data on the elderly, decentralized blockchain technology can be used to protect privacy rights [15].

### 2.2. Aquaponics Systems

Aquaponics refers to the novel combination of aquaculture and hydroponic culture [16]. The ecological cycle principle is illustrated in Figure 1. The cycle begins with the excrement (ammonia) discharged from aquatic animals, which is dissolved through aquaculture water. Then, a submerged motor pumps the aquaculture water to a filtration system to filter large excrement. Afterwards, the aquaculture water is flowed through a nitrification system [17]. The nitrification system contains a considerable number of nitrite bacteria, which can decompose ammonia and oxidize it into nitrates. In an anaerobic environment, it can be decomposed into nitrogen and oxygen to achieve nitrification. The aquaculture water is then drained into a plant-growing bed. As the plants' roots absorb nitrogen fertilizer, the water is purified [18]. Then, the pure water is sent back to the bucket

to rear aquatic animals, therefore not requiring large amounts of artificial fertilizers. There is no danger of pesticide residues, and the only requires regular maintenance of the entire aquaculture ecosystem to achieve a symbiotically productive environment for agriculture and fishery culture [18].

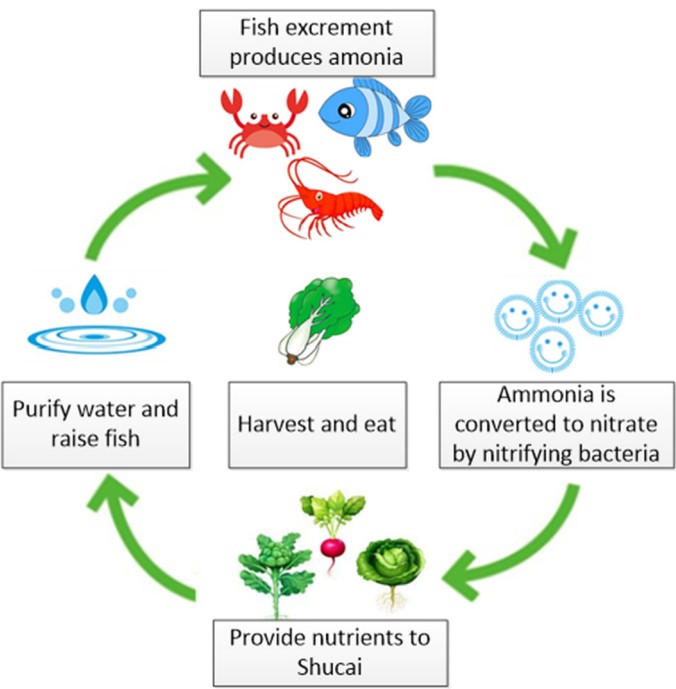

**Figure 1.** An aquaponics ecological cycle system.

Aquaponics provides the following advantages: (1) saving manpower, with no crop rotation barriers; (2) environmental protection and water saving; (3) self-sufficient agriculture fisheries culture systems; (4) non-toxic, safe, and clean; (5) rapid growth of agriculture and fishery; and (6) energy savings and carbon reduction; among other benefits. Aquaponics systems are ideal for metropolitan communities with limited land [19]. In addition to the problem of an aging population and declining birthrate, as the country develops gradually toward urbanization and industrialization, most of young and middle-aged people will choose urban employment rather than working in the more difficult agriculture or farming industries. Therefore, many scholars have suggested that agriculture and fisheries should be combined the Internet of Things (IoT) technologies and wireless transmission technology for agricultural and fishery production, such as the in the research cases presented in [5,6]. IoT technology and sensors can be used to sense the water quality and soil conditions in such systems. The collected data can be transmitted to a device designated by the user through wireless transmission technology, such as Wi-Fi, ZigBee, Bluetooth, etc., allowing users to monitoring or analyze collected data and then adjust environmental factors based on the results of the analysis to achieve more ideal growth conditions for aquatic animals and crops.

For the convenience of management, in most cases, most equipment will be remotely controlled. Therefore, a remote-control system is particularly important for user authentication. If remote access is not secure enough, the farming environment could be affected and seriously damaged. With respect the authentication of such Internet of Things systems, many studies have been conducted, with researchers proposing a variety of solutions [20–24].

### 2.3. Cloud Computing

Cloud computing, a new academic concept, is an Internet-based computing method that employs a considerable amount of equipment and software located in a large room

to provide applications through the sharing of resources. Through cloud computing, application services provide instant services and applications in various aspects through cloud computing. Due to the rapid development of technology and the popularization of Internet bandwidth and speed, the application services provided by cloud computing are now more applicable to all-round services, such as artificial intelligence and satellite navigation [25].

Cloud deployment can be broadly divided into public clouds, private clouds, and hybrid clouds. Public clouds are an open cloud service platforms provided through third-party cloud providers. Users can use application services through the Internet at a relatively low cost. Private clouds are the opposite of the public cloud, providing a single-enterprise or personal service. Cloud computing architecture can be built and managed by an individual or enterprise administrator with high security, scalability, and stability but with a relatively high cost to build. A hybrid cloud, as the name implies, includes the features of both public and private cloud computing, transferring non-sensitive information to the public cloud for computations and returning the results to the private cloud, thus preventing the private cloud from exceeding its computing performance [26,27].

Figure 2 illustrates the structure of software as a service (SaaS), platform as a service (PaaS), and infrastructure as a service (IaaS). The differences between these three services are outline as follows:

- Software as a Service (SaaS) is a software service model in which software and related data are centrally hosted in the cloud. Users can use the software without installing it on their local device, instead accessing software services via the Internet through a browser. Software as a service has become a common delivery model for business applications such as accounting systems, customer relationship management software, management information systems, enterprise resource planning, and human resources management.
- Platform as a service (PaaS) provides computing platforms and solution services. The PaaS layer falls between SaaS and IaaS, allowing users to deploy cloud infrastructure to the client to obtain services using programming languages, libraries, or other platforms. The user does not need to manage the network, server, operating system, or storage control; they only need to deploy an application environment.
- Infrastructure as a Service (IaaS) provides basic cloud resources, such as cloud computing units, storage units, and network components. Users can almost completely control cloud host resources. They can deploy and run processing, storage, network, and other basic computing resources at will without purchasing network infrastructure equipment, such as servers and software. Users cannot control the underlying infrastructure but can control the operating system, storage devices, and deployed applications. Sometimes, users also have limited control over certain network components, such as host-side firewalls.

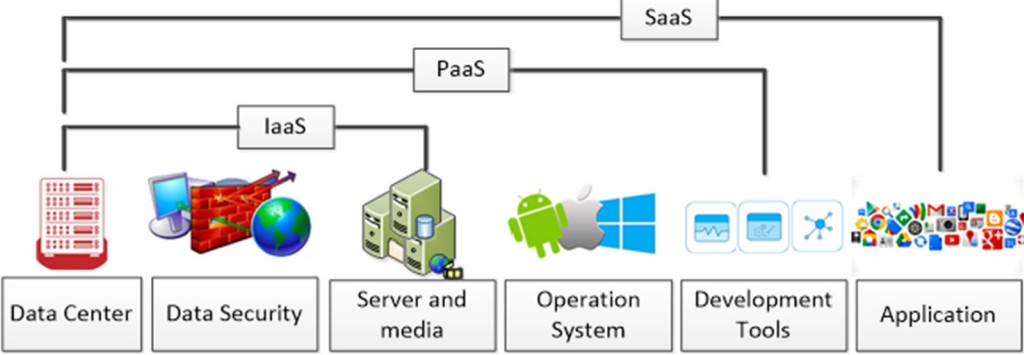

**Figure 2.** Three types of cloud computing services.

In addition, the Internet of Things environment includes a large number of sensors, through which a large amount of data is collected and transmitted. If the transmitted data must be analyzed and used in real time, it will face the problem of resource consumption through cloud computing. Therefore, edge computation [24] has been propose to efficiently solve these problems.

### 2.4. Internet of Things

The Internet of Things was proposed by Kevin Ashton in 1999 and by Germany as the basis for the development of Industry 4.0 in 2011. Its main goal is to enhance the intelligent digital development of traditional manufacturing [28]. Through real objects, such as vehicles, machines, appliances, etc., APIs or embedded sensors are connected to a variety of wireless or wired means of communication (item-to-object communication, item-to-person conversation, and people-to-person conversations) to provide management and service functionality. At present, the IoT combines artificial intelligence and big data to enable a wide range of applications, such as smart factories, smart homes, smart appliances, smart health care, and smart cities.

According to the 2005 International Telecommunication Union (ITU) report, the Internet of Things will connect things in the world in sensory and intelligent ways [29,30]. In the world of tomorrow, everything will be connected through the IoT, enabling direct communication with devices from anywhere. Thus, ITU describes the Internet of Things at four levels: item identification (markers), sensors and wireless sensor networks (feeling objects), embedded systems (thinking objects), and nanotechnology.

According to Cai et al. [31], the topology of the IoT is defined as follows: each node in the region handles different sensing devices and collects information relevant to cloud services; each node provider can correspond to one or more topologies. The IoT topology proposed by Wang et al. is divided into four layers: the application layer, cloud computational resource layer, middleware layer, and perception layer. The application layer can provide a wide range of Internet services. The cloud computational resource layer uses a demand program, task program, and task dispatcher procedures to meet the service quality requirements of each service demand to the greatest extent possible; at the most appropriate time, it can effectively allocate appropriate resources for each service demand. The middleware layer executes data processing and authentication, whereas the perception layer collects the sensor data required by the application layer.

The Internet of Things (IoT) refers only to identifiable things (objects) and their virtual representations in an Internet-like structure. The main advantage of the IoT concept is that it has a considerable the impact on many aspects of potential users' daily lives and behaviors. From the perspective of private users, the most obvious effects of the introduction of the IoT will be visible in the work and home fields. Currently, some industrial, standardization and research institutions are participating in the development of solutions to meet outstanding technical requirements [32].

The IoT is a network formed by connecting real-world objects (such as vehicles, machines, and household appliances) through APIs or embedded sensors linked by various wireless or wired communication devices to provide communication and dialogue for management and other services [33,34]. The IoT architecture is mainly divided into four layers, including a perception layer, network layer, IoT application support layer, and application layer [35]. The four-layer architecture of the IoT is shown in Figure 3.

The four main IoT architecture layers are described below:

- The perception layer is primarily responsible for data collection and transmission. This layer is divided into sensing and recognition technologies. Sensing technology enables connected objects to detect changes in the environment or the movement of objects via sensors. Identification technology mainly comprises radio frequency identification (RFID), a wireless communication technology. Radio signals can identify specific targets and read and write related data without the need to establish mechanical or optical contact between the identification system and specific targets.

- The network layer is primarily responsible for delivering messages recognized or collected in the perception layer to the application platform or application layer. Therefore, it can be called described as a bridge between the perception layer and the application layer. The network layer's main technologies are Wi-Fi, Bluetooth, ZigBee, TCP/IP, etc.
- The IoT application support layer receives the data transmitted by the perception layer through the network layer and uses the cloud or the Internet of Things application platform to process and store the data. It can be combined with big data analysis and other technologies. Analysis produces predictive results or decisions and delivers their results to the application layer for service or management functions.
- The application layer is the most important part of real-world IoT execution, using the perception layer, the network layer, and the IoT application support layer to provide various applications and services, which can be used in medical, transportation, agriculture, culture, food, and other industries. As Internet of Things technology matures, it is foreseeable that future development and applications will become increasingly diversified, making our lives more intelligent.

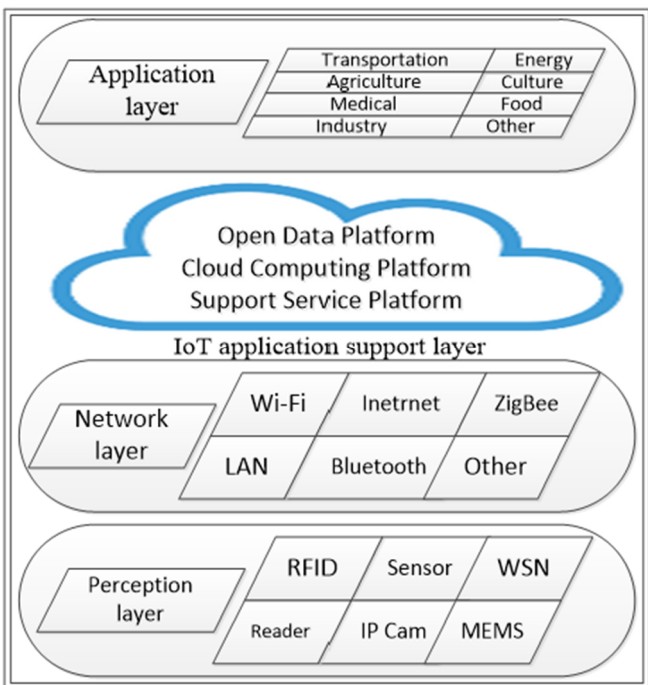

**Figure 3.** The four-layer architecture of the IoT.

The IoT has many applications in today's society. To build a secure IoT environment, privacy and data protection issues need to be considered. First, we must overcome network and data security issues. Therefore, many scholars have proposed network security solutions combined with the IoT [14,21,36,37], as well as solutions for data privacy and security [20,23,24,38–41].

*2.5. Green Care*

Green care is the concept of building an interactive environment through agriculture, animals, plants, gardens, and the landscape, allowing patients to interact with the environment to promote their physical and mental health [42], mainly for people suffering from psychological and mental illness, and providing a friendly healing environment. Green care is also known as care farms, ecotherapy, farm animal-assisted care, horticultural therapy, nature-based rehabilitation, nature-assisted therapy, or therapeutic gardens. Countries conducting leading scientific research in this field include the Netherlands, the UK, Norway, and Sweden [43], encouraging patients to reduce anxiety through experimenting with

nature, feeling nature, and connecting to nature to improve their quality of life [44–46]. Buist et al. [45] found that in addition to promoting the physical and mental health of patients, green care can also enhance patients' social activities and make them more willing to interact with the outside world, which can have a significant impact on patients' health and well-being. De Boer et al. [47] suggested that nursing staff should actively provide rehabilitation activities for patients, making full use of the green care environment so that patients can experience a warm and comfortable recuperation environment. Compared with traditional nursing homes, small home-based nursing homes are more beneficial to patients [47] and are more likely to stimulate patients' willingness to participate in rehabilitation activities and social interaction. Green care involves the integration of traditional healthcare systems with therapeutic environments, such as care farming, healing gardens, and ecotherapy [48]. At present, most traditional convalescent environments focus on the construction of environmental landscapes and are seldom combined with the medical care system to consider the various needs of patients, caregivers, and staff members. According to the abovementioned studies, the green care environment can reduce anxiety, promote physical and mental health, and improve the quality of life of patients. De Boer et al. [47,49] emphasized that under the current model of person-centered care, the convalescent environment tends to focus on small-scale homecare environments, adopting an open environment that is conducive to the flow and control of people, with the traditional medical model trending toward community care and a home care model; this mode is more effective than the traditional convalescent environment. De Boer et al. proposed that a green care environment should take into account the needs of patients, caregivers, and staff members, for example, by make rehabilitation and social activities accessible to patients with physical disabilities. Such a model can reduce the workload, even when caregiver manpower is limited, and maintain the green care environment in the most convenient way.

Based on the above research, in this study, we aim to strengthen current research by considering the needs of patients, caregivers, and staff members. The aim of this research is create a small-scale home care environment and ensure functions such promotion physical and mental health, encouraging rehabilitation and social activities, reducing the workload of caregivers, and automating environmental control. Furthermore, we aim to integrate the green care environment with the medical care system to improve the lack of traditional healthcare institutions and strengthen the functions of the traditional green care environment.

The system proposed in this study will be described below and compared with the traditional green care environment.

## 3. The Proposed System

In this section, the proposed system will be discussed in detail based on the concept of therapeutic environment and an IoT-enabled green care SGCS.

### 3.1. The Concept of the Proposed System

Minopoulos et al. proposed a medical system architecture that combines multiple technologies using a number of novel technologies [50]. In this paper focuses on the system architecture and data immediacy, with less focus on information security and communication security. We will refer to the medical system architecture proposed by Minopoulos et al. to strengthen communication information and data security.

The aim of the proposed system is provide urgently needed medical care for the elderly to prevent diseases, promote environmental treatment, and generate labor. To this end, we integrate IoT human–computer interaction through the concepts of therapeutic environment and green care. Such integration can effectively enhance the interaction between the elderly and the therapeutic environment through green care to provide physical care and rehabilitation to achieve physical health and mental well-being. The proposed IoT SGCS can perform real-time environmental monitoring in a therapeutic environment. For example, in most previous studies, automatic control of the monitored event was performed

to monitor water quality and soil in aquaponics. In addition to providing an automated control process for the system, in this study, we integrated urgent push notifications for monitoring events, whereby information is pushed to the phone of elderly patients, alerting them to get up and deal with the incident. The system also provides a standard for handling operation procedure (SOP) media, which allows users to easily respond to events. With respect to intelligent voice control, the proposed SGCS improves the intelligent voice control system (IVCS) [7] proposed by Wang et al. By using mobile phones as the carrier of the voice control system to achieve remote environment automatic control, the problem of the elderly not being able to operate information systems is considerably improved, also fulfilling the requirements of purpose of physical care, rehabilitation treatment, stress adjustment, health promotion, and improvement of physical and mental pleasure for the elderly. The integration of the proposed SGCS is shown in Figure 4.

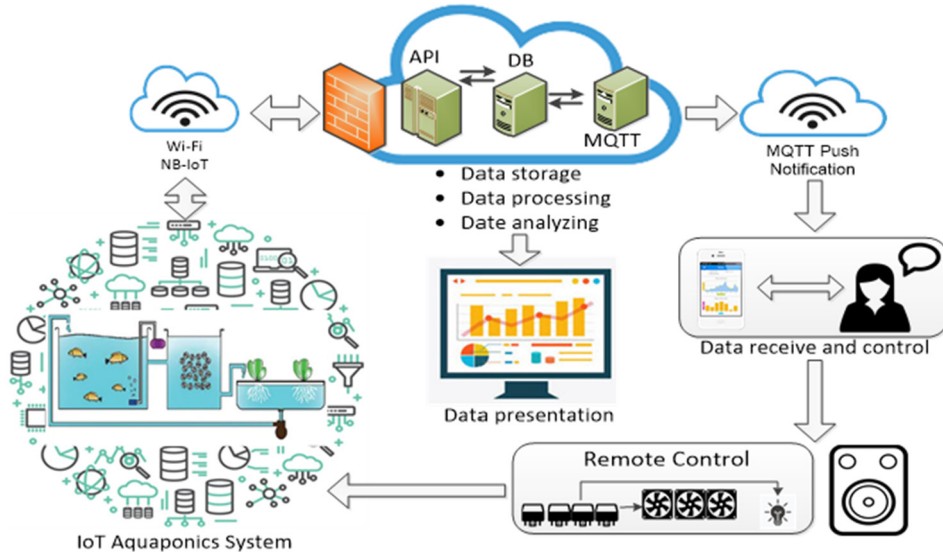

**Figure 4.** The integration of SGCS.

In this study, a fish tank system is combined with an aquaponics system to create a healing green care environment. IoT components are integrated to monitor the entire system environment. The IoT monitoring components include low water volume monitoring, water turbidity monitoring, PH value monitoring, water temperature monitoring, soil moisture monitoring, soil temperature monitoring, environmental temperature monitoring, and environmental humidity monitoring. A mobile app is provided for the users so they can stay up-to-date with the latest environmental analysis data and receive environmental event processing SOP. The back end consists of a message queue telemetry (MQTT) system that delivers push notification to the front-end system, providing real-time environmental analysis data through cloud environment analysis, which is presented on the electronic interface. In order to achieve the objectives of this study, we constructed a SGCS, the structure of which is described in detail below.

### 3.2. Security of SGCS

Alzahrani et al. [51] pointed out that the security of IoMT equipment includes the two categories of medical equipment and health information. The information security risk levels of these two categories should be distinguished, and varying degrees of risk control should be established. The risk control of medical equipment should be stricter and precise, mainly because if the medical equipment is hacked and the equipment is interrupted, the patient's life may be at risk. The fuzzy AHP-TOPSIS method proposed by alzahrani et al. can classify medical devices and health information, establishing varying degrees of risk control. Although the SGCS equipment described in this study is not medical equipment,

we used the MQTT server account, password and release MQTT Topic for security control, with health information protected with token key, timestamp, and API key signature.

The cloud server security, communication information security, push notification security, data packet format, and data packet verification procedures of the proposed SGCS are explained below.

- Security of Cloud Servers: SGCS cloud hosting includes a firewall server, API service server, database server, and MQTT server. All sensor data in the SGCS environment must call the RESTful API on the cloud for transmission, and only after filtering the approved IP through the firewall can the data be sent to the API service host to access the database host. Every RESTful API service on the cloud will be checked for communication security.

- Security of Communication Information and Data: The sensor data in the SGCS environment must call the RESTful API on the cloud through the Wi-Fi or NB-IoT communication protocol, and the transmitted data packet must contain a token key, call timestamp, API key signature, and data. The token key is the universally unique identifier (UUID) of the microcontroller unit (MCU) device, the call timestamp is the timestamp associated with a call to the RESTful API, and the API key signature is the result of SHA512 hashing with data after formatting as a string. Data are expressed in the JavaScript object notation (JSON) data interchange format, and the content contains the information to be transmitted. A salt key is the key agreed upon in advance between the cloud server and the edge device MCU. These data exist in the database server and edge device MCU and are mainly used to obfuscate the information during transmission to avoid being obtained by someone with malicious intentions through the packet capture program; then, the RESTful API is called on the cloud according to the packet.

- Security of Push Notifications: In this research, the MQTT communication protocol is used to send user message notifications. Any edge device connected to the MQTT server needs to provide an account and password for verification to ensure communication security. The release of different edge device messages will be distinguished according to the difference of MQTT Topic; for example, the format of Topic is $/SGCS/Device/1118$, which represents the edge device with a UUID of 1118 in the SGCS environment. In order ensure that the published message can be delivered to the edge device, the quality of service (QoS) of the message publication will be executed only once.

- Structure of the Data Packet: The data packet is transmitted in JSON data interchange format as follows:

{
*"Token_Key"* $= TK$,
*"Call_Timestamp"* $= TS$,
*"API_Key_Signature"* $= S$,
*"Data"* $= DC$
}

The format is standardized with reference to the principles of communication information and data security. The propose SGCS adopts JSON data interchange format as the packet format, mainly because this format has the advantages of high compatibility, easy reading and modification, and support for multiple data types.

- Pseudocode for Authenticating Data Packets: The proposed SGCS will package the data packets to be transmitted according to the information and data security specifications, then make an API call to the API server on the cloud and pass the relevant parameters. The data packet verification process is as follows:

$RESTfulAPI \rightarrow CheckSignature(TK, SK, TS, DC, S)$
$F \leftarrow StringFormat("ts = \{TS\}; d = \{DC\}; sk = \{SK\}")$
$E \leftarrow SHA512Encrypt(DC, F)$
$if: E = S \rightarrow True$
$else: \rightarrow False$

The parameters are token key, call timestamp, API key signature, and data in sequence, which will be formatted according to the string before verification. SHA512 hash calculation is performed on the formatted $F$ and $DC$. Finally, the hashed $E$ and $S$ are compared. If $E$ and $S$ are the same, the verification is successful; otherwise, the verification fails.

### 3.3. Smart Green Care System (SGCS)

To build an IoT SGCS based on the concepts of therapeutic environment and green care, the first task is to understand the concepts of therapeutic environment and green care. In this study, we conducted a survey of the elderly service management system of the National University of Science and Technology and integrated an aquarium fish tank system and aquaponics system in compliance with the concepts of therapeutic environment and green care. To relieve stress and promote physical and mental health, the aquaponics system includes fish farming and vegetable growing, which can initiate a discussion of breeding and planting among the elderly, allowing them to share their experiences and promote social activities. When the fish and vegetables are harvested, they can host a non-toxic and healthy feast. The get-together can help the elderly connect with each other. The proposed aquaponics system integrating fish tanks is shown in Figure 5 as planned; the system was built according to the design architecture.

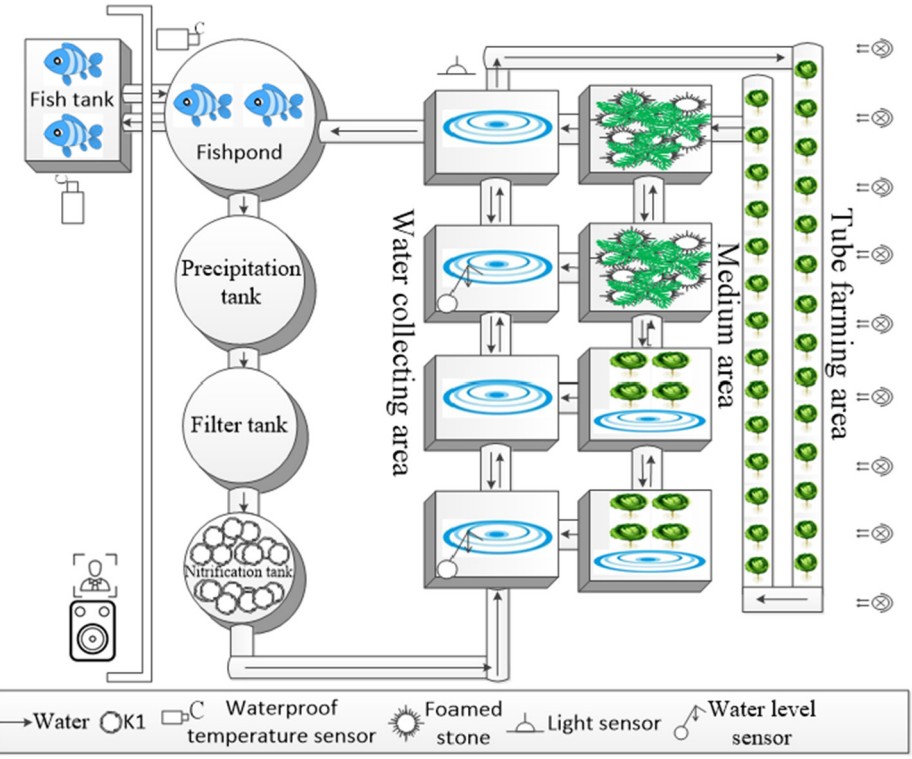

**Figure 5.** Integrated aquaponics system for fish tanks.

The main parts of the system consist of fish tanks, fishponds, a precipitation bucket, filter bucket, nitrification bucket, pipe farming areas, hydroponic areas, medium areas, and catchment areas. As shown in Figure 5, fishponds are mainly used for the cultivation of selected aquaculture, such as Wu Guo fish or Nile fish. The sediment barrels are mainly

used to precipitate all cultivation water returned through the aquaponics system. Because the water may contain unfiltered impurities, it will be precipitated during this process. After the cultivation water passes through the sedimentation barrels, it will flow through the filtration bucket. At this stage, the unfiltered impurities will be filtered with a finer filter material, and the cultivation water will be sent to the nitrification barrels. The role of the nitrification buckets is to decompose ammonia in fish excrement or residual feed. Because the nitrification system contains a considerable amount of nitrite bacteria, ammonia can be decomposed and oxidized into nitrate, which is then decomposed into nitrogen and oxygen by anaerobic bacteria. Then, the cultivation water is sent to the pipe cultivation area, the hydroponic area, and the medium area to allow the plants to absorb nitrogen fertilizer, completing the entire aquaponics cultivation system with circulating water.

A wide variety of environmental factors affects plant growth. Monitoring can enable the effective collection and analysis of planting environmental factors, helping to identify and improve the environmental factors that interfere with plant growth. To solve the problem of planting, in this study, we consulted with aquaponics operators to summarize the sensing equipment needed to monitor breeding and planting environmental factors (see Table 1 and Figure 6). The environmental monitoring system configures the sensing equipment in the aquaponics system, and the data collected by the sensing equipment provides the environmental data analysis model of the back-end system for analysis and processing. Compared with Gaussian NB, KNN, linear support vector classification (SVC), DT, AdaBoost, RF, and algorithms such as extra trees and gradient boost (GDB), the XG-Boost algorithm is superior in terms of classification and decision accuracy. The gradient boost decision tree (GBDT) algorithm is used to obtain higher accuracy with lower computing resources [52]. In this study, we used the XGBoosting decision tree model [53] for environmental data analysis and automatic control decision making, which is divided into a training phase and a testing phase. The following methods were employed:

- Training phase: The training data set is first processed for missing values, and each piece of data is marked. After the data is marked, the training program is implemented until it ends, at which point the weight of the trained model is returned and its parameters are stored for subsequent use.
- Testing phase: The input test data set is processed for missing values, and the trained mode is read. After inputting the test data set into the model, the model outputs the prediction result and returns the prediction result.

**Table 1.** List of IoT sensing devices.

| Device | Device |
|---|---|
| Water temperature sensor | Water quality pH sensor |
| Relative humidity sensor | Air temperature sensor |
| Soil moisture sensor | Raindrop sensor |
| Light sensor | Single-chip control board |
| Barometric pressure sensor | Wind speed and direction sensor |

The main purpose of the automatic control model is to simulate the maintenance experts of the aquaponics environment and to make decisions and control the aquaponics environment. Therefore, the model collects a large amount of data controlled by experts. The data fields of environmental monitoring data and automatic control will be labeled with the equipment by experts as the data labels, which will be used as the basis for building the automatic control model. The automatic control model will be built in the back-end server. This model monitors environmental data for automatic control and pushes event processing notifications to the front-end APP. Front-end users can immediately process the reported events on the spot or conduct maintenance through remote voice control to maintain the normal operation of the system. In addition to collecting environmental data through sensing elements, a small camera is used to capture images in the environment,

and the images are periodically sent to the server for identification through the established plant pathology model; the identification results are then pushed to the front-end APP.

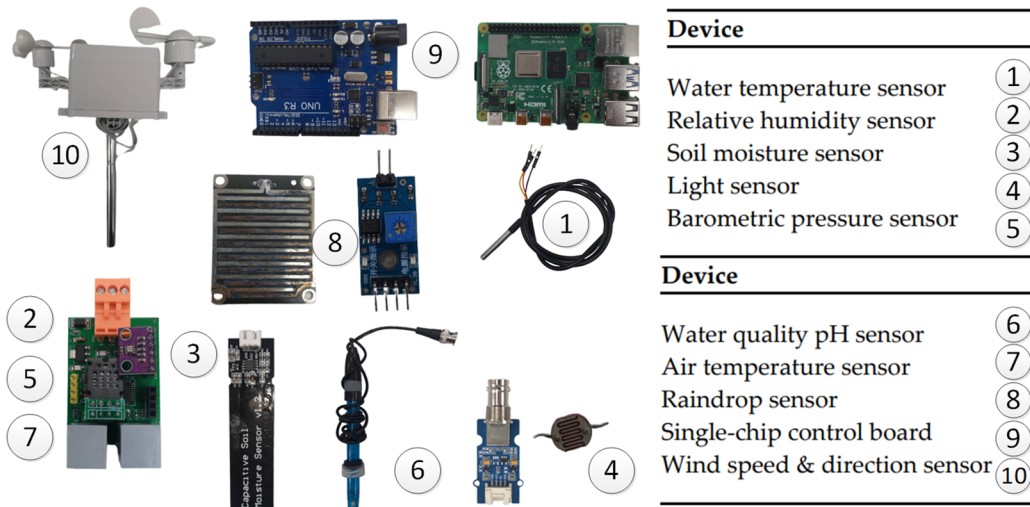

| Device | |
|---|---|
| Water temperature sensor | 1 |
| Relative humidity sensor | 2 |
| Soil moisture sensor | 3 |
| Light sensor | 4 |
| Barometric pressure sensor | 5 |

| Device | |
|---|---|
| Water quality pH sensor | 6 |
| Air temperature sensor | 7 |
| Raindrop sensor | 8 |
| Single-chip control board | 9 |
| Wind speed & direction sensor | 10 |

**Figure 6.** IoT sensing devices.

In this study, ResNet [54] is used as the phytopathology model architecture. This model was proposed by He et al., providing a very strong and stable ability to extract image features. Hsu et al. [55] also adopted this method for digital image recognition, achieving very good recognition accuracy. ResNet achieves satisfactory results in image classification, detection, and localization. Alghamdi et al. [56] applied ResNet to pain expression. Therefore, in this study, it is used as the core architecture of the phytopathology model, which is divided into a training phase and a testing phase. The following methods were employed:

- Training phase virtual code:

*Input* : *TrainingDataSets D*
*Output* : *Model*
*Set epoch*, *I*
*T* <= *TransfromToTensor(D)*
*L* <= *LabeledData(T)*
*M* <= *InitResNetModel()*
*Loss function* <= *Crossentropy()*
*Optimizer* <= *Adam()*
*For I* = 0; *I* < *epoch* ; *I* ++
*M'* <= *Training Process(M, T, L)*
*End*
*SaveModelParameters()*
*Return Model*

The *epoch* will be set as the number of model training phases, the input data set will be preprocessed and converted into the tensor *T* data format, and each data point will be marked as *L*. The trained model architecture is ResNet. After initializing the model weights, the *M* is returned, and the loss function and optimizer are set. At this stage, the trained model (*M'*) will be returned, and its parameters will be stored for subsequent use.

- Testing phase:

*Input* : *TestingDataSets D*
*Output* : *Predicted Results*
$T <= TransfromToTensor(D)$
$M <= LoadResNetModel()$
*Predicted* $R <= Testing Process(M, T)$
*Return Predicted Results*

The data for model testing will be converted into tensor $T$ data format after reading, and the trained ResNet model ($M$) will be read. After the test data are input into the model, the prediction result ($R$) will be returned.

The areas of the aquaponics system integrated in the fish tank are described below:

- Fish tank: The main purpose of setting up the fish tank is to incorporate the concept of therapeutic environment. The natural setting of the fish tank can create a feeling of relaxation and comfort. In this study, we set up a fish tank using cultivation water filtered by an aquaponics cycle. No additional filtration system is required, and an IoT water level sensor is used to automatically replenish water to minimize the cost of manual maintenance and to avoid the death of farmed fish due to human errors.

- Fishpond: The main function of fishpond is aquaculture; the proposed system integrates the cultivation of Wu Guo fish and Nile fish. During the breeding process, breeding waste and fish excrement will be produced, which can be used to provide nutrients in the water. However, excessive amounts of excrement will affect the quality of cultivation water and cause ecological disasters for fish and crops. Therefore, we will pump the water from the fishpond into the precipitation bucket to filter large impurities.

- Precipitation bucket: Through the cultivation water drawn from the fishpond, we will use large brushes to filter the breeding waste and excrement. The fine impurities that cannot be filtered are filtered twice by sedimentation. At this stage, a large amount of waste that affects poor water quality will be removed, and the cultivation water will be sent to the filter bucket for detailed filtering.

- Filter Buckets: This area is mainly used to filter finer feeding waste and excreta. Therefore, we use filter cotton and biochemical cotton for filtering. At this stage, the cultivation water will still contain small impurities, so it will be sent to the nitrification pond for digestive decomposition.

- Nitrification bucket: In this study, a large number of cultivation rings, biochemical balls, and K1 filter materials will be deployed in the nitrification bucket (please refer to Figure 7), and an air pump will be used to pump air to enable rolling of the cultivation ring, biochemical ball, and K1 filter material. Nitrifying bacteria will be cultivated during the rolling process. Micro-impurities in the cultivation water can be digested and decomposed by the nitrifying bacteria. This process will aid in the cultivation of plants, and the air pump will increase the oxygen content in the cultivation water. In addition to helping fish to breathe, it can also prevent the roots of plants from rotting.

- Pipe farming area: We will build a system of hydroponic plumbing area, which can be built vertically to save space. After precipitation, filtration, and nitrification, the cultivation water contains a considerable amount of nitrogen and oxygen, which will aid in plant nutrient absorption and hydroponic cultivation.

- Hydroponic area: The hydroponic area is cultivated in the same way as the pipe farming area but can provide a larger area of cultivation, which is more conducive to root growth.

- Medium area: We will build a medium planting area for the system because all plants are suitable for hydroponic cultivation, and medium can replace soil with the same effect without affecting cultivation water quality. Therefore, we will use volcanic rocks and foaming stones (please refer to Figure 8), sawdust, water moss, or coconut fibers as growth media. This area must allow the water level to produce a tidal effect, so

additional siphons must be added, which will aid in the respiration of plants and promote plant growth.

- Catchment area: The main function of the catchment area is to collect the cultivation water. The siphon device through the nitrification bucket and the medium area will collect the cultivation water into this area, which can also serve as a precipitation purification area. Water can be replenished from this area, and the cultivation water can be sent to the fishpond and the planting area to form a water circulation system to ensure effective use of water resources.

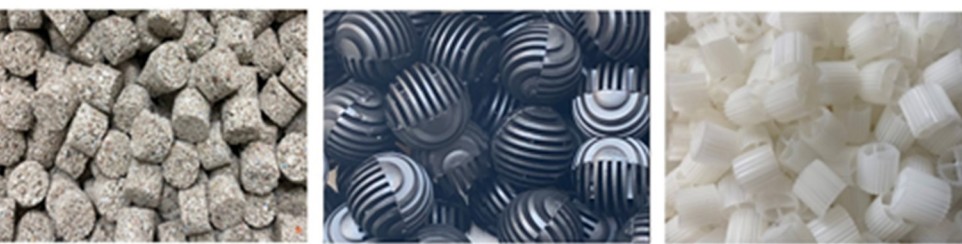

**Figure 7.** Cultivation ring, biochemical ball, and K1 filter material.

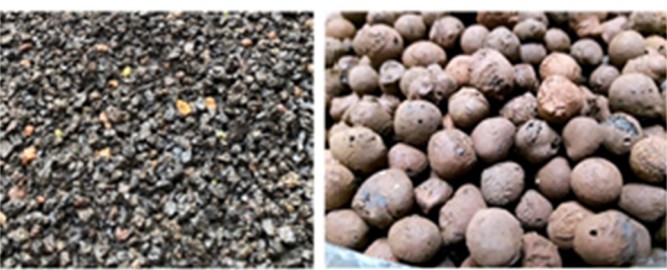

**Figure 8.** Fine stone and volcanic rock.

Figure 9 shows the constructed SGCS according to the system architecture presented in Figure 5.

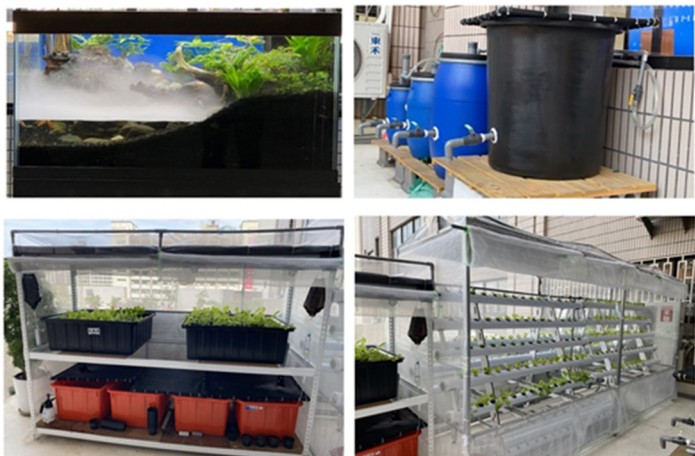

**Figure 9.** Smart green care system.

The SGCS proposed in this study includes a facial recognition and voice control system located between care areas and aquaponics systems, the main function of which is to provide caregivers with control over staff movements and to identify the elderly when they enter the therapeutic environment. Figure 10 shows a schematic diagram of the face recognition system, which is integrated with the intelligent voice control system (IVCS) proposed by Wang et al. [7]. The voice control interface allows the elderly to talk to the APP provided by this study to control the environment through the core of the cloud

voice system. Because identification has been carried out, human–computer interaction with the elderly is possible, with records documenting the interactions. The records are integrated into the activity history of the personal care resume. These activity history records provide the medical team and the nurses with a physical fitness analysis as a reference for rehabilitation care.

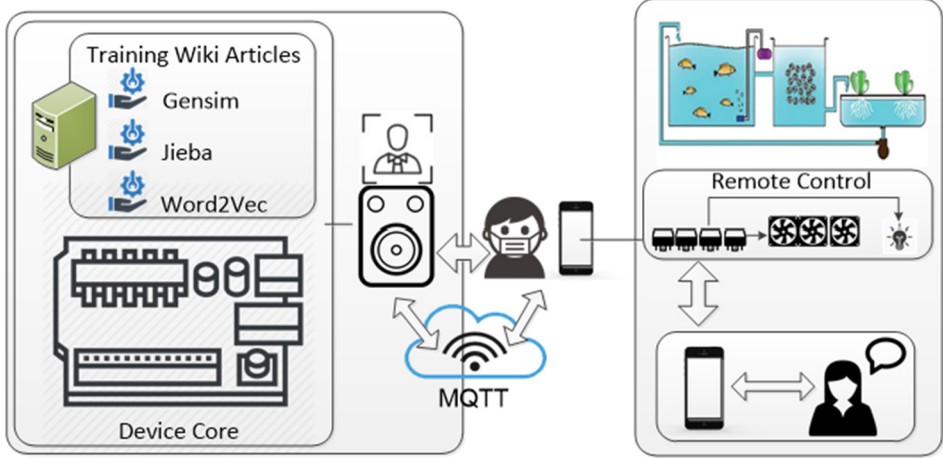

**Figure 10.** Face recognition and voice control system.

Wang et al. proposed IVCS [7], which uses the Gensim service to extract the titles and content of three million Wiki articles; then, the Jieba service to hyphenates the sorted titles and content. In the process of hyphenation, stop words are introduced to filter out slanderous words. Finally, the Word2Vec service is used for word vector training. After training, a core word vector similarity comparison model is obtained. IVCS publishes environment control commands through the cloud MQTT server. Because the published information will specify the UUID of the receiving device, the MCU device with the same UUID will receive the push notification and parse the control commands in the push information, satisfying user remote control requirements.

The proposed SGCS includes a set of event notification apps used by senior citizens. The main function is to push the monitoring environment information in the aquaponics system. Elderly people can use this app to explore all IoT sensing devices, with key data include air temperature sensors, relative humidity sensors, air quality PM2.5 sensors, soil temperature and humidity sensors, water temperature sensors, water PH detectors, water nitrite concentration value (NO2) detectors, water ammonia concentration value detectors, water oxygen concentration value detectors, and low-water hydration sensors. The SGCS will transmit these data to the back-end system through the RESTful API, and the back-end system will store, process, and analyze the data and push the results to the front-end APP and display on the screen (please refer to Figure 11). The caregiver app can present pending event notifications and completed work events, whereas the elderly APP can instantly monitor all the analysis results of the collected data. When the back-end system discovers an urgent event that needs to be handled, events and SOP files are pushed to the responsible seniors, as shown in Figure 11, at which point the senior citizens receives a notification alerting them of the need to feed the fish. Step 1 reminds seniors to get the food. Step 2 directs the seniors to the event handling area and teaches them the steps of feeding, which can be performed in person or via voice control from their phones. Step 3 informs them of successful feeding and records processed events. Interacting with the elderly through the APP and promoting the physical rehabilitation of the elderly can effectively enhance the pleasure of work, activate thinking, generate social topics, encourage them to enter the therapeutic environment, and relieve life stress.

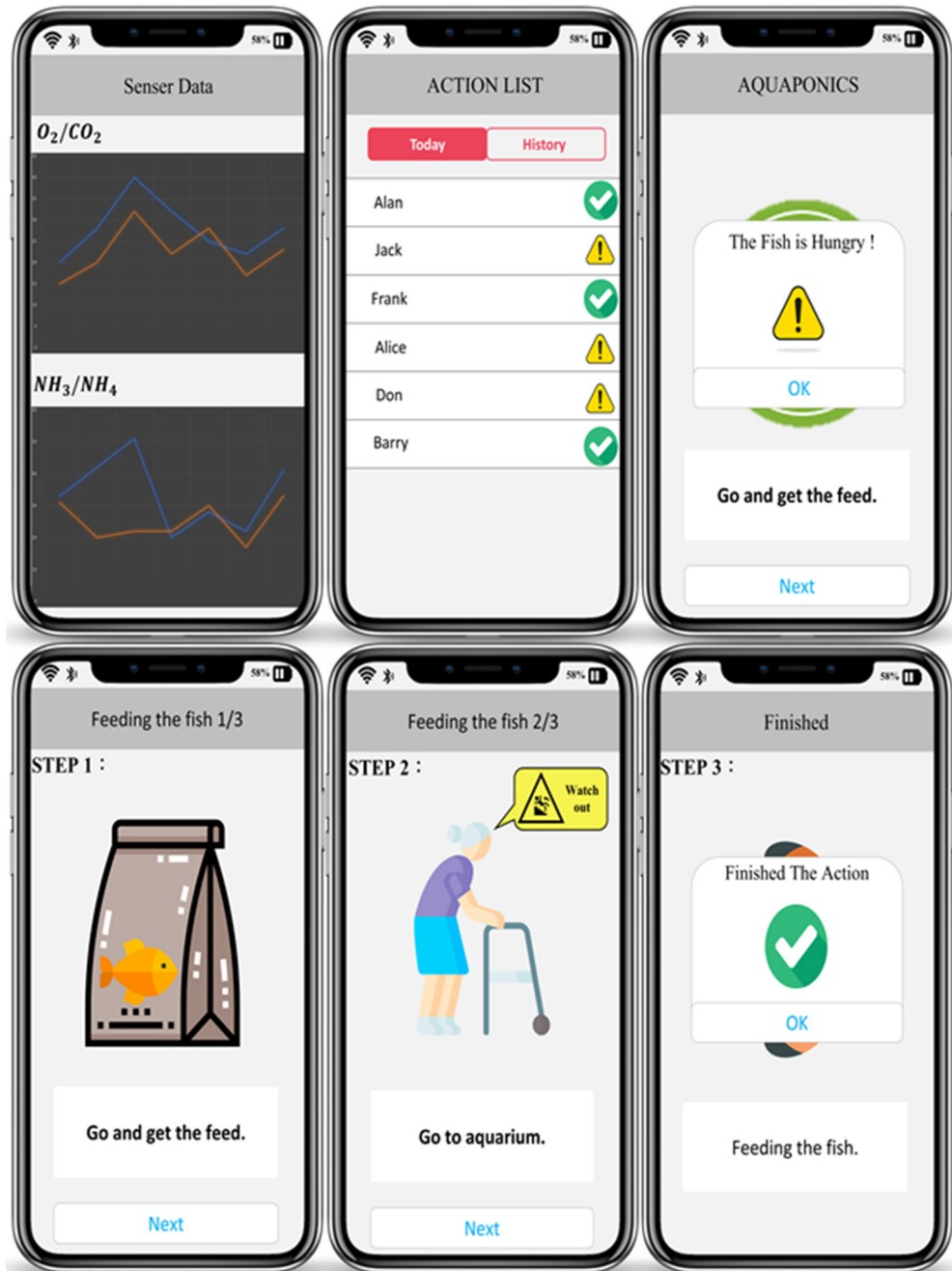

**Figure 11.** SGCS app.

Figure 12 shows the SGCS voice control system architecture. A large situation screen will be displayed in the care area, showing all IoT sensing device data that have been received, as well as the pending personnel event processing notifications. The caregiver or senior sends a communication command to the voice control system after face recognition when the event is not completed within the expected time frame; for example: start fish feeding or activate water rehydration, etc. The voice core system can automatically control the aquaponics environment and complete the required event processing and notification work. The work of caregivers to maintain the SGCS can be reduced through this framework.

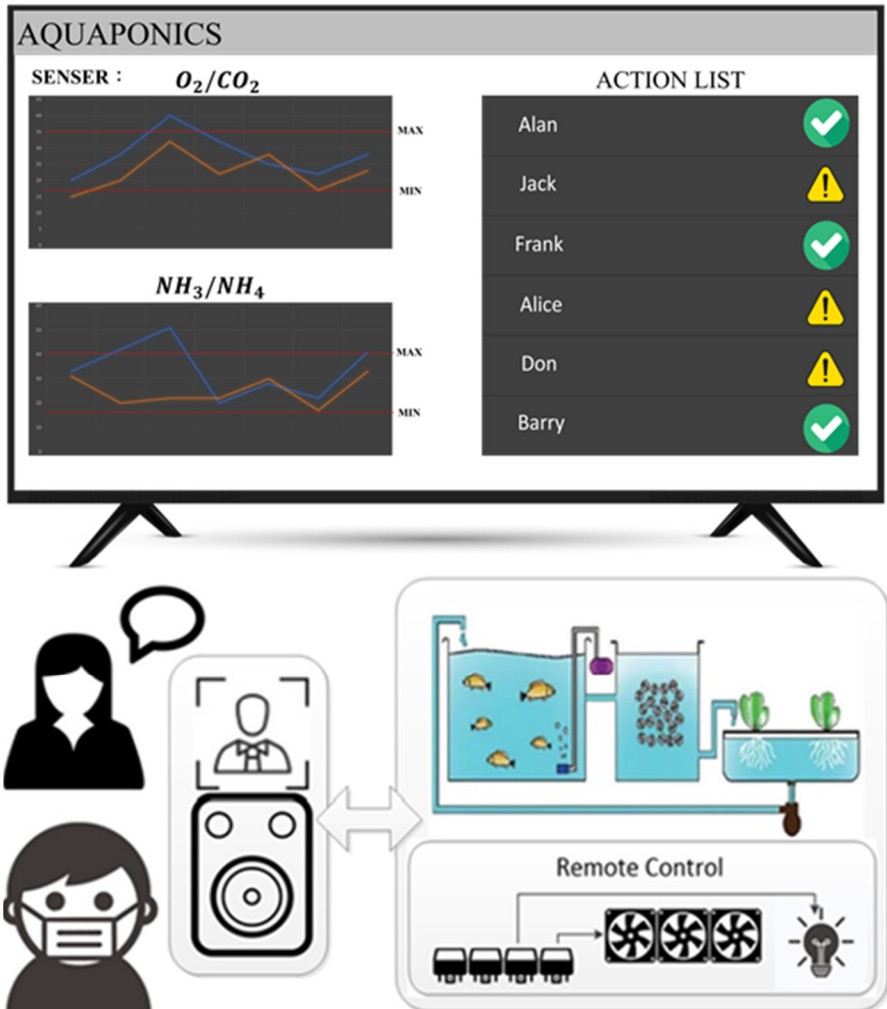

**Figure 12.** Voice control system.

Figure 13 shows the software service architecture of the SGCS, which includes three parts: an for the elderly and caregivers, an electronic whiteboard system, and a cloud system. The software services and functions included in the three systems are described below:

- App for the elderly and caregivers: This app comprises software functions, such as presentation of IoT sensing device data, event handling operation SOP, maintenance notifications, personal activity history, and voice control.
- Electronic whiteboard system: This system includes software functions, such as presentation of IoT sensing device data, information on the follow of elderly patients [50], and voice control.
- Cloud system: The cloud system includes the MQTT push service, data analysis service, event processing SOP delivery service, and other software services.

In this study, we verified the PoC, PoS, and PoB for various software functions and services to verify the feasibility of the SGCS technology services and business operations. Moreover, the proposed system fully implements the concept functions of "therapeutic environment" and "green care", making it suitable for various environments, such as long-term care institutions, medical units, and home care.

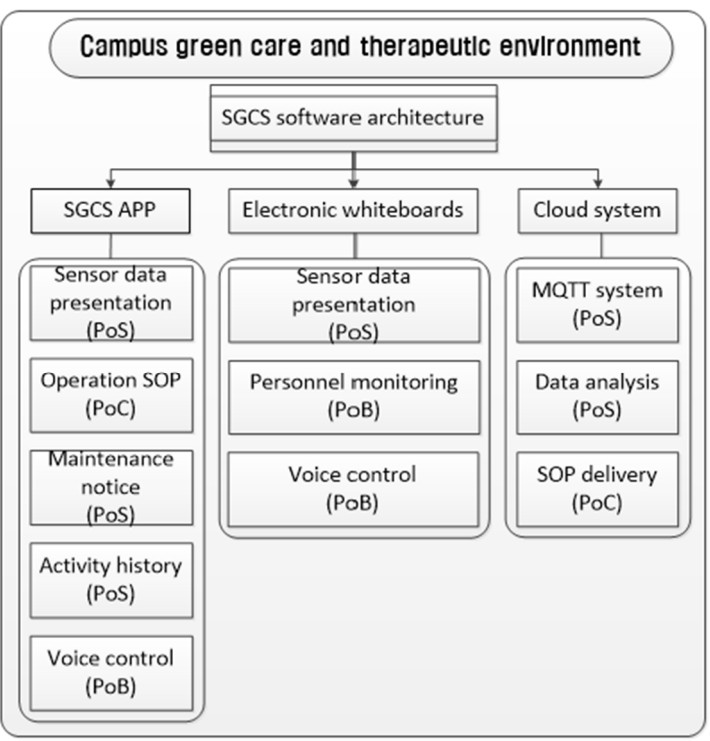

**Figure 13.** SGCS system architecture diagram.

### 4. Discussion and Analysis of the Effectiveness of the Proposed SGCS

In this section, we analyze and discuss the feasibility of technology services and business operations of the proposed SGCS. Table 2 lists analysis and discussion items, including manpower, cost, energy saving, safety, contamination, restriction, and rehabilitation care issues. The table shows that using the SGCS to build a healing green care environment is more effective than the traditional approach. The proposed SGCS can be used to improve the physical and mental comfort and quality of life of the elderly, in addition to providing stress relief and physical rehabilitation. The proposed software and hardware system architecture can be effectively implemented regardless of technology, service, and business operation mode.

In this study, we proposed an automatic control system using the Xgboost architecture to build an analysis model. The data used to build the analysis model were collected from the experimental environment and labeled by aquaponic experts. The data were labeled using four control devices: aquatic weed lamps, pumping motors, rain covers, and drip sprinklers. Example are presented in Table 3.

Table 3 shows real data collected from the experimental environment in this study. Experimental environment data will be collected every day to inform the automatic control model. In this study, we collected a total of three months of data from environmental sensors at 5-min intervals every day, which were recorded and marked according to the status of each switchgear at the time of data collection. A total of 28,800 pieces of data were collected in this study. During construction of the automatic control model, the training set and the test set were divided in a ratio of 8:2. The training set was used to build the model for testing and to evaluate the accuracy of the model. The automatic control system proposed in this study controls many devices by inputting experimental environmental data through the analysis model. Therefore, it is important to analyze whether the model can accurately predict the equipment control status. The prediction accuracy of the four device states is shown in Table 4.

**Table 2.** Comparison of traditional therapeutic environment green care with the proposed SGCS therapeutic environment green care.

| | Traditional Environment and Green Care | |
|---|---|---|
| | **Traditional Therapeutic Environment Green Care** | **SGCS Therapeutic Environment Green Care** |
| Manpower | The maintenance, monitoring, and control of environmental conditions must be performed in person to provide effective physical care for the elderly. | The introduction of remote monitoring and real-time warning can reduce manpower and arrange physical rehabilitation for individual seniors. |
| Environmental protection and energy saving | Environmental maintenance through rules of thumb is less grounded and more resource-intensive. | Referencing the data returned from IoT sensors to achieve the maintain the environment; maintenance resources can be used efficiently. |
| Food security | The use of pesticides to eliminate pests and diseases cannot be avoided. | The use of drugs is strictly prohibited by providing green nutrients through the ecological cycle. |
| Production quality | The crops produced are susceptible to imbalances in supply and demand, as well as to ecological imbalances. | Ecological advantages of precise farming production and easy-to-control plant growth. |
| Maintenance cost | Maintenance of the environment requires considerable human and resources. | Eco-cyclic IoT fields need to be built initially, and the environment can be monitored with alerts ensuring effective control of people and resources. |
| Land pollution | The use of organic or chemical agents is required for soil conservation and water recycling. | The use of off-ground farming and medium instead of soil can effectively reduce contamination. |
| Site restrictions | A large area is needed for green plant farming, with a limited land area and limited space use. | Effective planning can be implemented for the site; the SGCS can be large or small and can be developed horizontally or vertically to effectively use space. |
| Promote social activities | Only passive waiting for seniors to communicate and discuss the environment. | An app is provided to convey farming information, guide participants to online or offline communication topics, and effectively promote social activities. |
| Therapeutic care | Only passive waiting for the elderly to carry out physical fitness activities. | The system can provide physical rehabilitation for the elderly, as well as effective physical therapy. |
| Traceability | None | Environmental parameters can be stored, and data can be tracked. |
| Remote monitoring | None | The server pushes messages to the client side, enabling a remote monitoring environment. On the client side, the human voice can be used to control the environment through the server. |
| Real-time alerts | None | Alert thresholds can be set for changing environmental conditions. |
| Human–machine interaction | None | An alter app is provided to send notifications about maintenance processes and guide users to engage in physical fitness and care activities according to SOP. |

Table 4 shows that the average prediction accuracy of the automatic control model established in this study is as high as 96.8%. Based on the accuracy rate, the control model can accurately predict the functions of experts in maintaining various systems of the aquaponics environment. In addition to the accuracy evaluation to confirm that the automatic control model can accurately control the device, there is no overfitting problem, so the automatic control model prediction was evaluated using a confusion matrix. The confusion matrix of the automatic control model prediction Device04 is shown in Figure 14.

**Table 3.** Experimentally collected environmental field data.

| Class | Data |
|---|---|
| Temperature A (°C) | 28.9 |
| Temperature B (°C) | 28.2 |
| Humidity A (%) | 76.8 |
| Humidity B (%) | 85.5 |
| Photometric (lm) | 0 |
| $CO_2$ (ppm) | 433.0 |
| Soil Temperature (°C) | 28.4 |
| Soil Humidity (%) | 38.4 |
| Soil EC (mmhos/cm) | 0.14 |
| Soil pH | 8.9 |
| Wind Speed (m/s) | 0.0 |
| Wind Direction (°) | 164 |
| Rainfall Detection | 0 |

**Table 4.** Prediction accuracy for each device.

| Device Num. | Accuracy (%) |
|---|---|
| 01 | 99.7 |
| 02 | 94.6 |
| 03 | 94.9 |
| 04 | 98.1 |

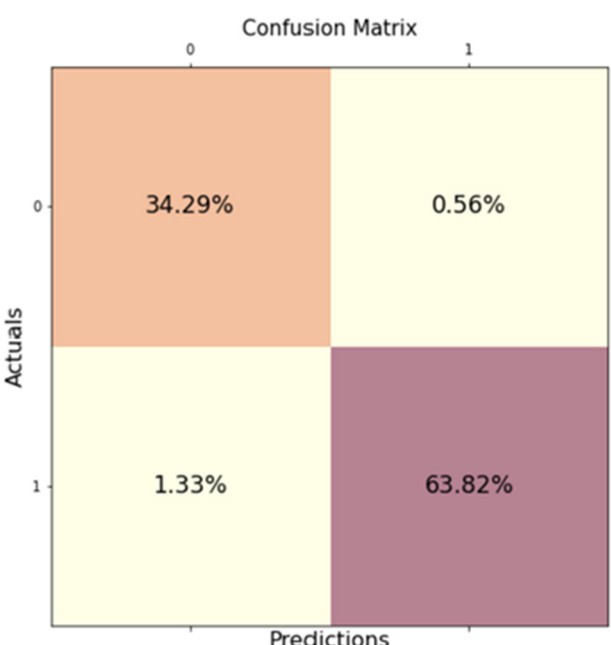

**Figure 14.** Confusion matrix for automatic control model prediction results.

The confusion matrix shows that the automatic control model does not suffer from overfitting and can accurately predict the two types of data. The device marked as 0 represents the open state, devices marked as 1 represents the closed state, and the recall value labeled as 0 is as high as 98.11%. To further evaluate the automatic control model, the behavior of Device04 is predicted by analyzing the automatic control model; the importance of each data field is shown in Figure 15.

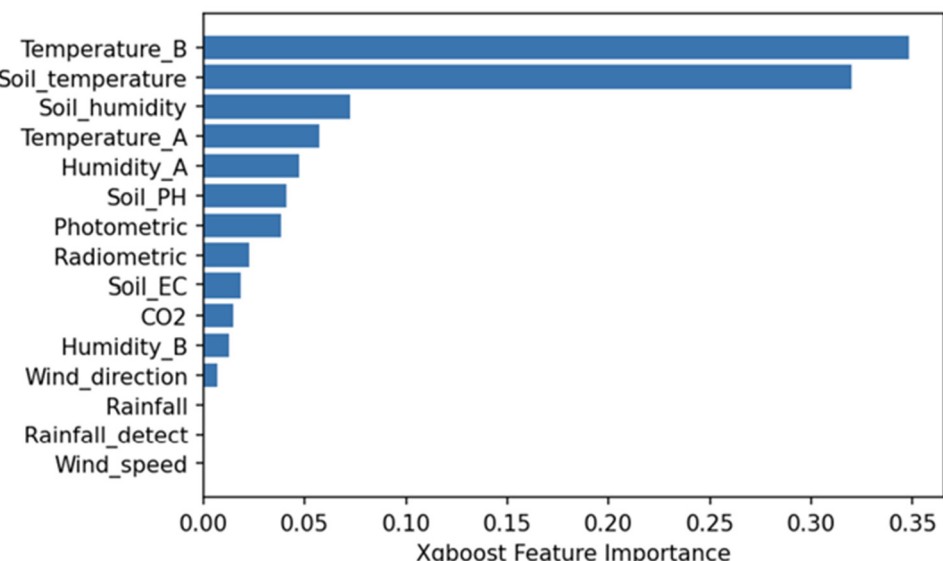

**Figure 15.** Importance of each data field with respect to the prediction results.

As shown in Figure 15, when the automatic control model predicts the state of the equipment, it will take the temperature B field as the main reference; the value and data in this field are shown in Figure 16.

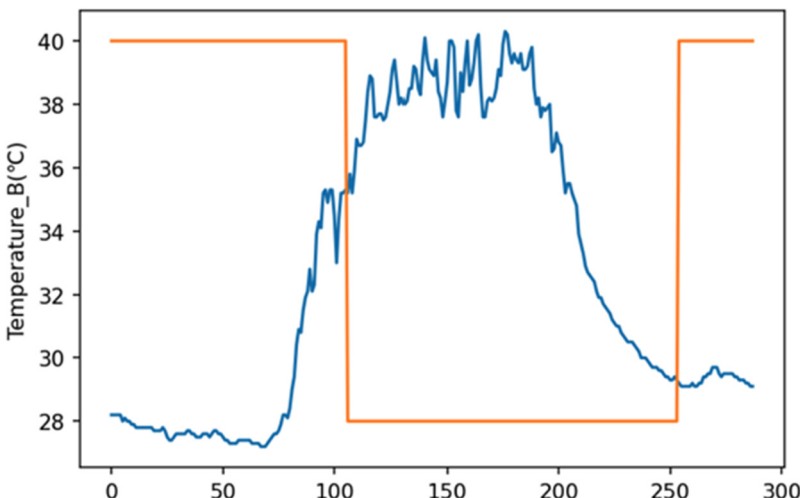

**Figure 16.** The importance of Device04's in the predicted analysis model.

The figure shows the data for a complete day; the blue line represents the air temperature data, the orange line represents the marked data, and an index of 0 represents the data at 00:00 in the morning. To facilitate viewing of the data, the marked values 0 and 1 represent two states, i.e., off and on, and the numerical value is enlarged to a value of 40, representing the on state, with a value of 28 representing the off state. The results show that the device is set to be on when the sun is not rising, and it is turned off when the sun is high in the sky, around noon. Open source leaf pathology data from the PlantVillage data set [57] was used to train the leaf pathology detection model. This function mainly uses the Tomato mosaic virus and Tomato healthy categories to build the pathology detection model. An example of the data set is shown in Figure 17a,b.

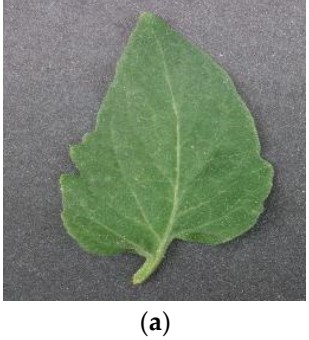 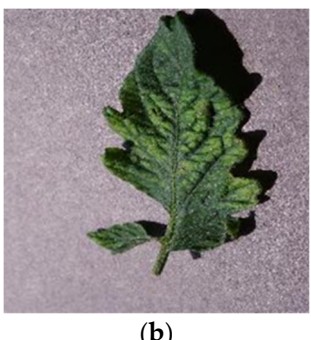

(**a**)    (**b**)

**Figure 17.** Sample of the PlantVillage data set: (**a**) healthy leaf sample; (**b**) mosaic virus leaf sample.

The amount of data in the two categories is shown in Table 5, in which the training set data has 1866 images and the testing set has 100 images.

**Table 5.** The number of training and test set details.

| Class | Training Set | Testing Set |
|---|---|---|
| Mosaic virus | 1542 | 50 |
| Healthy | 324 | 50 |

The architecture used in this study to build the leaf pathological detection model is ResNet [54]. The model achieves high accuracy and stability in terms of image recognition, so this architecture was used to build the leaf pathological detection model. After the model was built through the training data set, the model performance was verified through the test data set; the results are shown in Figure 18. The model uses Adam [58] as the optimization algorithm. The hyperparameters are set as follows: batch size, 32; learning rate, 0.001; and training epoch is set to 30. After training, the performance of the model was verified through the test data set; the results are shown in Figure 18.

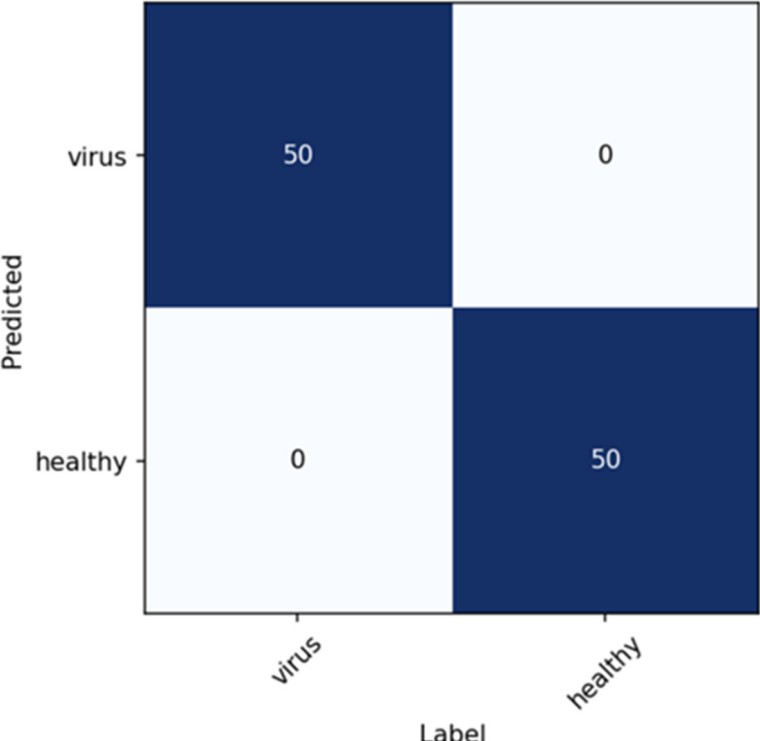

**Figure 18.** Confusion matrix of model identification results.

The *x*-axis in Figure 18 represents the data label, and the *y*-axis represents the prediction result of the model. When the prediction result is equal to the label, the data are correctly classified. This figure shows the leaf pathological identification model constructed in this study in the form of a confusion matrix, where the prediction of the data set and the number of corresponding markers is tested. The results show that the leaf pathological identification model established in this study achieved an accuracy rate of 100% on the test data set. This model should be used to assist in identifying the health status of tomatoes grown in the experimental environment to achieve accurate identification results.

## 5. Conclusions

The SGCS proposed in this study realizes and satisfies the concept of a healing environment and green care. The system uses human–computer interaction to encourage care recipients to perform physical rehabilitation. The care and rehabilitation records generated by the system can be used by long-term care institutions, as well as units such as medical institutions and home care institutions, and meet the needs of caregivers to provide daily care records for care recipients. For system maintenance and operation problems, the SGCS uses a decision tree model to analyze environmental data, providing automatic control and decision making via remote equipment, adopting a phytopathological analysis model, conducting automatic monitoring of plant cutting, actively issuing early warnings about plant cutting disease, and using automatic management to reduce maintenance costs. In response to personnel needs, the proposed SGCS uses a voice control system to assist physically handicapped elderly patients in participating in care activities, a face recognition system to assist in the management and control of people flow, assists caregivers in managing care work through electronic whiteboards, and uses information technology to reduce personnel cost. With respect to system security issues, the proposed SGCS combines token keys, timestamps, and hash signatures to ensure communication security and cloud data security. In view of the cost problem, the proposed SGCS makes is applicable in homes and institutions as it reduces maintenance and operation costs, reducing personnel costs and using a cloud service architecture design. The proposed SGCS can be implemented in care institutions provided by the Department of Management of Campus Elderly Services, using PoC, PoS, and PoB to verify the feasibility of the technology, services and business operations support, and recognition and satisfaction to promote the physical and mental health of the elderly and to relieve the pressure of caregivers in a timely manner.

Unlike traditional green care environments, the SGCS proposed in this study is well-suited for community and home care environments, in addition to integrating green care environments with healthcare systems. The needs of patients, caregivers, and staff members are also considered in order to address the lack of traditional convalescent institutions and strengthen the functions of traditional green care environments.

The proposed SGCS still needs to consume electricity to care for the environment. In the future, it can be combined with solar energy or recycled water resources to generate green energy to reduce costs, in addition to mitigating disasters caused by power outages. For some data that needs to be analyzed and processed in real time, edge computing can also be strengthened to maintain the performance of cloud servers. For confidential and sensitive data, blockchain and information security technology can be integrated to achieve decentralized personal data protection. At present, there is a lack of personnel danger detection and notification function in the proposed environment, which can also be improved by using intelligent assistive devices or imaging technology.

**Author Contributions:** W.-L.L. and S.-C.W. conceived the proposed method; W.-L.L. and L.-S.C. performed the experiments; W.-L.L. and L.-S.C. developed the software; L.-S.C. and T.-L.L. analyzed the data; W.-L.L. and S.-C.W. wrote the paper; J.-L.L. was responsible for project administration; T.-L.L. verified the system. All authors have read and agreed to the published version of the manuscript.

**Funding:** This research received no external funding.

**Institutional Review Board Statement:** Not applicable.

**Informed Consent Statement:** Not applicable.

**Data Availability Statement:** Not applicable.

**Conflicts of Interest:** The authors declare no conflict of interest. The founding sponsors had no role in the design of the study; in the collection, analyses, or interpretation of data; in the writing of the manuscript, and in the decision to publish the results.

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
