# Peer review of "Green Care Achievement Based on Aquaponics Combined with Human–Computer Interaction"

_applsci, doi:10.3390/app12199809_

Round 1

Reviewer 1 Report

This paper built a human-machine interactive APP which called Smart Green Care System (SGCS), that can take full advantage of the concept of “therapeutic environment” and “green care”. In addition, it combines an automatic control systems and intelligent voice control applications. It is different from the general therapeutic environment or the green care space. In addition, this study takes long-term care institutions as an example to verify the Proof of Concept (PoC), Proof of Service (PoS) and Proof of Business (PoB). After verification, it can be determined that the SGCS is indeed feasible and has the ability to spread. Therefore, it can be widely used in the field of long-term care in the future

In a satisfactory manner, the basic purpose of the research has been described, but with some crucial comments that should be taken into consideration.

1.     The abstract is very long and needs to be shorter.

2.     At the end of Section 1, INTRODUCTION, it is preferred to add a paragraph representing the contributions in this work "The main contributions in this study are: ......".

3.     In Figure 7 which represents the Integrated aquaponics system for fish tank, the author should remove the legend that represents the shapes of Water, K1, …, Plant from the top of the figure to become in a box in the bottom of the figure.

4.     Figure 8 shows the IoT sensing devices do not have the names of each element/device included in them.

5.     The confusion matrix presented in Figure 15 which shows the automatic control model prediction Device04 is presenting the value in numbers but we advise the author to replace the numbers 3224, 53, 125, and 6001 with the percentage equivalent value.

6.     In the title of Table 3, replace the word Collcection with the word Collection.

7.     In Table 3 which shows the Experimental Environment Collection Data Field some classes do not have their units like CO2, Soil_EC, Soil_PH, Wind_Speed, Wind_Direction, and Rainfall_detect.

8.     Figures 15, 16, 17, and 19 are unclear; the author should replace them with a higher-quality image.

9.     There are NO references from the years 2021 and 2022 in the list of references. We recommend that the author include at least SIX current references that are from the years 2021 and 2022 in order to improve the article. 

      In the list of references, references [1, 2, 3, 8, and 9] the author wrote “[Last accessed 10 and 11 September 2020]” please modify it to be a recent date. 

    Future work section is needed after/with the conclusion section

Author Response

We have revised our manuscript in response to the  referee’s comments and are currently asking a native English-speaking colleague to review the manuscript. 

Reviewer 2 Report

The paper does not include enough evidence to support the claim. The following bullet points include suggestions to improve the manuscript.

·         The introduction should be divided into some paragraphs according to the general pattern, research background, related works, challenges and contributions in the field of image security.

·         In general, details of all figures are missing.

·         Authors are advised to enhance the quality (size and resolution) of Figures.

·         Authors should provide introduction and commentary on related works for each proposed system.

·         It would enhance the understanding of the general reader if the system model given in is elaborated. because it is not clear what the meaning of the curve is.

·         The technical quality of this paper is quite low. Although theoretical concepts and related literature are properly introduced, some of the references are outdated and should be omitted. Moreover, the number of recent literature is low

·         Consider the following studies in the reference section:

 Comparative analysis of low discrepancy sequence-based initialization approaches using population-based algorithms for solving the global optimization problems

·         The authors need to follow the journal format for citing the references. References are incomplete.

·         Firstly,, authors should provide more specific comments of the cited papers after introducing each relevant work. What readers require is, by convinced literature review, to understand the clear thinking/consideration why the proposed approach can reach more convinced results. This is the very contribution from authors. In addition, authors also should provide more sufficient critical literature review to indicate the drawbacks of existed approaches, then, well define the main stream of research direction, how did those previous studies perform? Employ which methodologies? Which problem still requires to be solved? Why is the proposed approach suitable to be used to solve the critical problem? It needs more convinced literature reviews to indicate clearly the state-of-the-art development.

·         There are grammatical mistakes in the manuscript, the authors need to proofread it from native English speaker. Authors should use clearer and concise vocabulary to express their ideas and discussions.

·         that the paper still lacks the validation for the results which is very critical to check the contribution of this work compared to the current state of arts.

·         Related work section is very short. I suggest adding some more comparison of existing methods.In this study, it is necessary to make a comparison with the existing methods in the literature. If necessary, additional experimental studies should be added and the results should be shared in order to make comparisons.
Conclusion needs reconsideration. It needs to highlight more the research main contribution, briefly answering questions
'how ?' with some untrivial indications and improvement percentages to keep with the reader. Also, the conclusion needs to present some more ideas of open research and future work for researchers to build upon for further advancements.

1.      Optimization of neural network using improved bat algorithm for data classification

2.      Comparative analysis of low discrepancy sequence-based initialization approaches using population-based algorithms for solving the global optimization problems

3.      A fine-tuned BERT-based transfer learning approach for text classification

4.      Comparative research directions of population initialization techniques using PSO algorithm

Author Response

(The authors gave the same response as above.)

Reviewer 3 Report

The subject of research and content of the manuscript are with relevant and updated information. The results can be reproduced and open a valuable field of research.

The authors must expand the discussion section with information on the security of the data and information of the system because when verifying the effectiveness of the Smart Green Care System (SGCS) they must ensure its accessibility, robustness and reliability.

Author Response

(The authors gave the same response as above.)

Round 2

Reviewer 1 Report

I believe the manuscript has been sufficiently improved to warrant publication in Applied Sciences. 

Author Response

Reviewer 1:

I believe the manuscript has been sufficiently improved to warrant publication in Applied Sciences.

Ans:

Thank you for your kindness to read our paper. Also, thank you for accepting our manuscript.

Reviewer 2 Report

·         I do apologize in saying that the paper still lacks the validation for the results which is very critical to check the contribution of this work compared to the current state of arts.

·         Related work section is very short. I suggest adding some more comparison of existing methods. In this study, it is necessary to make a comparison with the existing methods in the literature. If necessary, additional experimental studies should be added and the results should be shared in order to make comparisons.
Conclusion needs reconsideration. It needs to highlight more the research main contribution, briefly answering questions
'how ?' with some untrivial indications and improvement percentages to keep with the reader. Also, the conclusion needs to present some more ideas of open research and future work for researchers to build upon for further advancements.

·         What readers require is, by convinced literature review, to understand the clear thinking/consideration why the proposed approach can reach more convinced results. This is the very contribution from authors. In addition, authors also should provide more sufficient critical literature review to indicate the drawbacks of existed approaches, then, well define the main stream of research direction, how did those previous studies perform? Employ which methodologies? Which problem still requires to be solved? Why is the proposed approach suitable to be used to solve the critical problem? We need more convinced literature reviews to indicate clearly the state-of-the-art development.

·          

Author Response

Reviewer 2:

Related work section is very short. I suggest adding some more comparison of existing methods. In this study, it is necessary to make a comparison with the existing methods in the literature.

Ans:

Thanks to your suggestion, more literature has been added in related work (see Section 2.5 for details). In our study, recent related research results have been analyzed and the shortcomings of these methods have been identified, and then improved methods have been proposed in our study. Besides, to highlight the contribution of this study, the results of our study are compared with traditional methods in Section 4. (Table 2. The Comparisons of Traditional Therapeutic Environment Green Care and SGCS Therapeutic Environment Green Care.) â–Š

  1. García-Llorente, M.; Rubio-Olivar, R.; Gutierrez-Briceño, I. Farming for Life Quality and Sustainability: A Literature Review of Green Care Research Trends in Europe.  J. Environ. Res. Public Health2018, 15, 1282.
  2. Steigen, A.M.; Kogstad, R.; Hummelvoll, J.K. Green Care services in the Nordic countries: an integrative literature review.European Journal of Social Work 2016, 692-715.
  3. Moriggi, A.; Soini, K.; Bock, B.B.; Roep, D. Caring in, for, and with Nature: An Integrative Framework to Understand Green Care Practices. Sustainability2020, 12, 3361.
  4. De Boer, B.; Beerens, H.C.; Katterbach, M.A.; Viduka, M.; Willemse, B.M.; Verbeek, H. The Physical Environment of Nursing Homes for People with Dementia: Traditional Nursing Homes, Small-Scale Living Facilities, and Green Care Farms. Healthcare2018, 6, 137.
  5. Buist, Y.; Verbeek, H.; De Boer, B.; De Bruin, S. Innovating dementia care; implementing characteristics of green care farms in other long-term care settings. International Psychogeriatrics 2018, 30(7), 1057-1068.
  6. Haubenhofer, D.K.; Elings, M.; Hassink, J.; Hine, R.E. The Development of Green Care in Western European Countries. EXPLORE 2010, 6, 106-111.
  7. De Boer, B.; Beerens, H.C.; Katterbach, M.A.; Viduka, M.; Willemse, B.M.; Verbeek, H. The Physical Environment of Nursing Homes for People with Dementia: Traditional Nursing Homes, Small-Scale Living Facilities, and Green Care Farms. Healthcare2018, 6, 137.
  8. Ura, C.; Okamura, T.; Yamazaki, S.; Shimmei, M.; Torishima, K.; Eboshida, A.; Kawamuro, Y. Rice farming care as a novel method of green care farm in East Asian context: an implementation research. BMC Geriatr2021, 21, 237.
  9. Bram de Boer; Jan, P.H.; Hamers; Sandra, M.G.; Zwakhalen; Frans, E.S.; Tan; Hanneke C.; Beerens; Verbeek, H. Green Care Farms as Innovative Nursing Homes, Promoting Activities and Social Interaction for People With Dementia. Journal of the American Medical Directors Association 2017, 18, 40-46.

If necessary, additional experimental studies should be added and the results should be shared in order to make comparisons.

Ans:

Thanks for your advice. Since our research is mainly focused on green care, a comparison of our research environment with the traditional green care environment has been added to Section 4. (see Table 2 in Section 4. The Comparisons of Traditional Therapeutic Environment Green Care and SGCS Therapeutic Environment Green Care.) â–Š

Conclusion needs reconsideration. It needs to highlight more the research main contribution.

Ans:

Thanks for your suggestion, the main contribution of this study has been added to the Conclusion section, and how to improve the deficiencies of traditional green care environment has also been explained. â–Š

The conclusion needs to present some more ideas of open research and future work for researchers to build upon for further advancements.

Ans:

Thanks for your suggestion, ideas for future work have been added at the end of the manuscript (see Section 5, Conclusions). â–Š

What readers require is, by convinced literature review, to understand the clear thinking/consideration why the proposed approach can reach more convinced results. This is the very contribution from authors.

Ans:

Thanks for your suggestion to our manuscript, more literature has been added in related work (see Section 2.5 Green Care). In Section 2.5, it is explained that the current green care environment does not take into account the needs of patients, caregivers and staff, and methods to meet the basic needs of green care have been developed, such as promoting health and encouraging participation in rehabilitation and social activities, etc. In our study, an approach based on considering the needs of patients, caregivers and staff for future work in the current literature has been proposed, such as:  encouraging patients to participate in rehabilitation and social activities through the system (SGCS APP), using voice warfare control system to relieve caregiver's work stress, and use decision-making of remote control  and phytopathological models for automated analysis to reduce the pressure on staff member to maintain a green nursing environment. â–Š

Authors also should provide more sufficient critical literature review to indicate the drawbacks of existed approaches, then, well define the main stream of research direction, how did those previous studies perform? Employ which methodologies? Which problem still requires to be solved? Why is the proposed approach suitable to be used to solve the critical problem?

Ans:

Thanks for your advice, more literature has been added to the related work (see Section 2.5), the lack of previous research has been addressed by our paper, and the solution to the problem has been added in our research. Since our research is combined with many current technologies and applied to green care environment, the lack of traditional green care environment can be eliminated. A comparison of our research environment with the traditional green care environment has been added to Section 4. (see Table 2 in Section 4. The Comparisons of Traditional Therapeutic Environment Green Care and SGCS Therapeutic Environment Green Care.) â–Š
